# Mechanisms Underlying the Rarity of Skeletal Muscle Cancers

**DOI:** 10.3390/ijms25126480

**Published:** 2024-06-12

**Authors:** David S. Kump

**Affiliations:** Department of Biological Sciences, Winston-Salem State University, 601 Martin Luther King Jr. Dr., Winston-Salem, NC 27110, USA; kumpds@wssu.edu; Tel.: +1-336-750-8782

**Keywords:** cancer, epidemiology, immune, incidence, metastasis, microenvironment, myokine, prevention, skeletal muscle

## Abstract

Skeletal muscle (SKM), despite comprising ~40% of body mass, rarely manifests cancer. This review explores the mechanisms that help to explain this rarity, including unique SKM architecture and function, which prohibits the development of new cancer as well as negates potential metastasis to SKM. SKM also presents a unique immune environment that may magnify the anti-tumorigenic effect. Moreover, the SKM microenvironment manifests characteristics such as decreased extracellular matrix stiffness and altered lactic acid, pH, and oxygen levels that may interfere with tumor development. SKM also secretes anti-tumorigenic myokines and other molecules. Collectively, these mechanisms help account for the rarity of SKM cancer.

## 1. Introduction

Skeletal muscle (SKM) comprises approximately 40–50% of body weight, depending on leanness [1,2,3]; nonetheless, primary SKM cancer is extremely rare [4,5,6,7], especially when organ mass is considered (Table 1). When rare primary tumors occur in SKM, they most often derive from soft tissues during growth in childhood (particularly early childhood [age ≤10 years]) and include rhabdomyosarcoma and rhabdomyoma [8]. Rhabdomyosarcoma is recognized by light and electron microscopy as a tumor harboring cells that resemble myoblasts [9,10]. Despite the similarity with myoblasts, it is unclear whether rhabdomyosarcoma arises from myoblasts [11]; nonetheless, myoblasts likely undergo high proliferation during early childhood and adolescence to support SKM growth [12] and therefore could be the origin of the tumor. From 1975 to 1999, the incidence rate of rhabdomyosarcoma in the U.S. was 4.5/1 million for individuals aged <20 years [13], which is similar to the rate of 4.9/1 million in Sweden in 2016 for individuals aged <15 years [14]; the rate is lower in reported Asian populations [15], perhaps suggesting a genetic contribution. In the U.S., SKM cancers account for ~3.5% of pediatric cancers [16].

In adults, the incidence rate of rhabdomyosarcoma is estimated to be ~9.6/10 million. This rate was derived by multiplying the soft tissue cancer incidence rate of 3.2/100,000 [17] by 3%, which is the fraction of soft tissue sarcomas that rhabdomyosarcoma comprises [18]—the actual number is likely much lower, as the incidence rate is not age-specific, and thus inclusion of the pediatric population may skew the results higher; moreover, one website [19] lists the soft tissue sarcoma-to-rhabdomyosarcoma fraction in adults at 1%—if this is accurate, the incidence rate would be at least three times lower. Moreover, when the organ mass is considered (incidence rate per 100,000 per g of tissue), rhabdomyosarcoma manifests by far the lowest values among all cancers, at just 0.005%, 0.18%, 0.03%, 0.01%, and 0.0002% of the values for breast, brain, kidney, pancreas, and prostate cancers, respectively (Table 1). Heart cancer has the second-lowest value but still manifests a 24-fold greater incidence rate per tissue mass compared to that of rhabdomyosarcoma.

**Table 1 ijms-25-06480-t001:** Comparison of incidence rates of cancer by organ and organ mass.

Cancer Type	Incidence Rate per 100,000 Individuals [Reference]	Incidence Rate Relative to Breast Cancer ^a^	Mean Organ Mass (g) [Reference]	Incidence Rate per Tissue Mass (per 100,000 per g Tissue) ^b^	Incidence Rate per Tissue Mass Relative to Breast Cancer ^a^	Number of Times More Common Than SKM Cancer Relative to Tissue Mass ^c^	Notes
Urinary bladder	5.6 [20]	0.120	37 [21,22]	0.151	1.56	30,545	Mean organ mass was obtained by averaging the values for men and women.
Bone	0.9 [23]	0.019	3465 [24,25]	2.6 × 10^−4^	2.7 × 10^−3^	53	Reported bone mass was multiplied by 0.33, the organic mass fraction.
Brain	3.5 [20]	0.075	1294 [26,27,28]	2.7 × 10^−3^	0.028	548	Mean organ mass was obtained by averaging the values for men and women from two reports each.
Breast	46.8 [20]	1	484 [29]	0.097	1	19,581	Organ mass indicates mean values for women only, although a high variation is present.
Colorectal	17.8 [20]	0.380	1818 [30]	9.8 × 10^−3^	0.101	1982	Incidence rate was derived by adding the rates of colon and rectal cancers, which were reported separately.
Gallbladder	1.2 [20]	0.026	16.4 [31,32]	0.073	0.757	14,817	
Heart	0.034 [33]	7.3 × 10^−4^	288 [34,35]	1.2 × 10^−4^	1.2 × 10^−3^	24	Mean organ mass was obtained by averaging the values for men and women.
Kidney	4.4 [20]	0.094	287 [27,28]	0.015	0.159	3110	Mean organ mass was obtained by averaging the values for men and women.
Leukocyte	13.7 [20]	0.292	1200 [36]	0.011	0.118	2303	Incidence rate was derived by adding the rates of Hodgkin lymphoma, non-Hodgkin lymphoma, multiple myeloma, and leukemia, which were reported separately. Organ mass includes the total mass of all immune cells.
Liver	8.6 [20]	0.184	1425 [27,28]	6.0 × 10^−3^	0.062	1223	Mean organ mass was obtained by averaging the values for men and women.
Lung	23.6 [20]	0.504	370 [27,28]	0.064	0.660	12,925	Mean organ mass was obtained by averaging the values for men and women.
Ovary	6.7 [20]	0.143	6.3 [37]	1.06	11.0	215,357	Mean organ mass was derived by multiplying the reported mean ovarian volume (6.3 mL) by the ovarian tissue density (1.00 g/mL).
Pancreas	4.7 [20]	0.100	91.8 [38]	0.051	0.529	10,368	
Prostate	29.4 [20]	0.628	11 [39]	2.67	27.6	541,227	
SKM	0.096	0.002	19,440 [40]	4.9 × 10^−6^	5.1 × 10^−5^	1	Incidence rate was calculated as explained in Section 1. Mean organ mass was derived by multiplying 0.44 (the approximate lower end of SKM mass as a fraction of total body mass [1]) by mean body mass (women and men averaged) [40].
Skin	13.6 [20]	0.291	3250 [41]	4.2 × 10^−3^	0.043	847	Incidence rate for skin cancer was derived by adding the rates of melanoma and non-melanoma, which were reported separately.
Stomach	9.2 [20]	0.197	142 [42]	0.065	0.671	13,129	Mean organ mass was obtained by averaging the values for men and women.
Testes	1.7 [20]	0.036	36.6 [43]	0.046	0.480	9401	Organ mass data were obtained in a population of men aged 41–50 years from northwest India.
Thyroid	9.1 [20]	0.194	16 [44]	0.553	5.72	112,021	Mean organ mass was obtained by averaging the values for women aged 20–69 years and men aged 30–69 years.
Uterus	22.5 [20]	0.481	515 [45]	0.044	0.452	8849	Incidence rate for uterine cancers was derived by adding the rates of cervix uteri and corpus uteri cancers, which were reported separately.

Abbreviations: SKM, skeletal muscle. Footnotes: ^a^ As breast cancer is the most frequent cancer type, it is used as a reference. ^b^ Calculated by dividing the incidence rate by the organ mass. ^c^ Calculated by dividing the incidence rate per tissue mass by that of SKM.

Secondary SKM cancer is also rare. For instance, in a 2015 single-institution retrospective study [4], SKM metastases were present in 31/1805 (1.7%) patients with cancer who underwent F-fluorodeoxyglucose positron emission tomography or computed tomography. The frequency of SKM metastasis varied by primary tumor site, ranging from 2.8% in lung cancer to 6.9% in melanoma. Isolated SKM metastases were uncommon (6.5%), and most were accompanied by other metastatic sites. Moreover, using computed tomography, a 2010 study detected SKM metastases in just 61/5170 (1.2%) patients with metastasized cancer [46]; metastases were most commonly associated with primary tumors of the genitals (24.6%), gastrointestinal tract (21.3%), urinary system (16.4%), and melanocytes (13.1%). Another study reported that SKM metastasis occurred in <1% of patients with kidney cancer [47]. In addition, a 2012 study [5] confirmed the rarity of SKM metastases in patients with lung cancer (1.6%), with most metastases occurring in trunk muscles, suggesting that SKM metastases likely arise from adjacent tumors. In patients with breast cancer, SKM metastasis typically suggests a higher-stage cancer [7], and in patients with non-small cell lung cancer and other metastatic carcinomas, SKM metastasis predicts a poor prognosis [48,49]. This may be because SKM metastases tend to appear late during the course of metastatic cancer [50], indicating that the disease is already widespread before it infiltrates SKM. As posited by LaBan et al. [51], SKM “is in fact a hostile environment for tumor emboli”.

The reasons underlying the relative rarity of primary and secondary SKM cancers remain obscure, but their infrequent occurrence may be due to characteristics unique to SKM architecture and function, the SKM microenvironment, and SKM-derived secretions such as myokines, some of which exhibit anti-tumorigenic properties. (Note that for the purposes of this review, “tumorigenic” refers to all attributes and steps involved in tumor development, progression, and metastasis.) This review examines and summarizes these postulated potential mechanisms, including the speculative involvement of specific SKM myofiber types. Improved understanding regarding these mechanisms could lead to the development of novel therapeutic approaches exploiting the anti-tumorigenic properties of SKM.

## 2. Epidemiological and Mechanistic Studies

A higher SKM mass, often measured using the SKM index, is correlated with improved prognosis in patients with cancers of various origins, including biliary tract, brain, breast, esophageal, gastric, liver, small-cell lung, ovarian, pancreatic, urothelial, and pediatric malignant solid cancers [52,53,54,55,56,57,58,59]. For example, a 2010 meta-analysis demonstrated that lower muscle mass was associated with an all-cause mortality hazard ratio of 1.44 in patients with solid cancers [60], although the effect is likely at least partially due to better resistance to chemotherapy-induced dysfunction supported by increased SKM mass. Nevertheless, a higher SKM index can intrinsically improve cancer outcomes [56,61].

Moreover, regular muscle contractile activity (i.e., exercise) helps to prevent cancer as well as enhance cancer treatment and improve patient prognosis [62,63,64,65]. For example, regular exercise or physical activity mitigates the risk of breast, bladder, brain, bone, colorectal, endometrial, esophageal, gastric, kidney, lung, and other cancers, often via humoral factors [62,63,66,67,68,69,70,71,72,73,74,75,76,77,78,79,80,81,82,83,84,85,86]. However, these epidemiological data only explain how muscular activity per se can impact other cancers rather than explaining the rarity of SKM cancers. Notwithstanding, insights gained from mechanistic studies examining this effect may also offer understanding regarding the rarity of SKM cancers. For instance, many of the proposed mechanisms outlined below, such as altered cytokines, anti-tumorigenic secretions, and shear stress, may be magnified in the local SKM microenvironment.

## 3. Local Mechanisms Alleviating Cancer in Skeletal Muscle

Mechanisms that may inhibit primary and secondary cancers in SKM include the unique characteristics and architecture of SKM, immune-related factors, the properties of the SKM microenvironment, and local myokine concentrations. These mechanisms are illustrated in Figure 1 and Figure 2.

### 3.1. Skeletal Muscle Characteristics and Architecture

#### 3.1.1. Amitotic Nature of Skeletal Myofibers

Terminally differentiated SKM myofibers are generally considered to be amitotic [87]. Although some evidence indicates the existence of DNA endoreplication without cell division in murine SKM [88], the development of tumors from amitotic tissue is uncommon [89,90]. For instance, in the nervous system, tumors arising from amitotic neurons are rare, with most tumors arising from glial cells [91]. Similarly, despite the unclear origin of primary SKM cancers, the most common forms, such as rhabdomyosarcoma and rhabdomyoma, appear to arise from cells such as myoblasts rather than from mature SKM myofibers [11]. Accordingly, the amitotic nature of SKM helps to explain the scarcity of primary cancer in SKM.

#### 3.1.2. Physical Barrier

The dense and organized structure of SKM provides a physical barrier that can impede the invasion and spread of cancer cells, thus helping to limit SKM metastases. For instance, SKM is composed of parallel myofibers that run the entire length of the muscle. Individual SKM myofibers are enveloped by a basement membrane, which is covered superficially by the endomysium, a collagen-rich connective tissue. Endomysium-covered SKM myofibers are bundled into fascicles that are surrounded by the perimysium, which is thicker than the endomysium. All fascicles in an individual SKM are encased by the epimysium, which is thicker than the perimysium. The collagen fibers of the endomysium, perimysium, and epimysium merge at the myotendinous junction and with the collagen fibers of the tendon [1,92]. This arrangement of interconnected SKM connective tissue provides a physical barrier to the migration and invasion of tumor cells from neighboring tissues. However, when metastases do occur in SKM, most are found between these different connective tissue layers [93,94,95]; accordingly, it has also been postulated that these connective tissues may serve as a guide for invading cells [96].

#### 3.1.3. Mechanical Forces

SKM undergoes frequent contraction and relaxation, which generates mechanical forces that could disrupt the adhesion and survival of cancer cells. Because the contractile myofilaments are physically connected to the extracellular matrix (ECM) via numerous proteins, the mechanical forces initiated by cross-bridge cycling and subsequent sarcomere shortening are transferred to the surrounding connective tissues, thus helping to transfer force more efficiently to tendons [97,98], creating shear forces within the contracting muscle [99], and therefore interfering with the ability of tumor cells to adhere to the ECM surface. The mechanical forces may also dislodge cancer cells that may have adhered to the intramuscular endothelium, thus preventing extravasation and infiltration of SKM [51]. Primary heart cancer is also rare (incidence rate, 34/100 million) [33]; secondary heart cancer is approximately 20- to 40-fold more common than primary heart cancer, with an incidence rate of ~8.5–17/100,000 [100]. The low incidence of heart cancer may also be related to the continuous, cyclic mechanical forces exerted by autorhythmic contractions of cardiomyocytes; this may be even more inhibitory to cancer development than occurs in SKM.

#### 3.1.4. Low Resting Metabolic Rate

Although energy metabolism in SKM can increase up to 100-fold during contractile activity [101], resting SKM has a low metabolic rate [102]. As SKM is typically in a resting state, this low metabolic rate may limit the occurrence of SKM cancers. This mechanism is explained by the Warburg effect, which describes the link between higher glycolysis and lactate production in cancer cells, despite oxygen availability [103]. This helps to describe the inverse association of cancer risk and outcomes with whole-body basal metabolic rates [104], although it could also be applied to individual organs [104]; thus, organs with higher metabolic rates may be at higher risk of cancer, whereas organs with lower metabolic rates may be at lower risk.

As tissue metabolism drives local blood flow and perfusion [105], an increased risk of cancer in organs with a higher metabolic rate may be related to increased glucose delivery. The risk may also result from increased tissue production of reactive oxygen species (ROS) as a byproduct of greater energy metabolism [106]: ROS can induce oncogenic damage to DNA and thus increase the risk of cancer in the local tissue [107]; ROS may also diffuse to tumoral cells and foster the epithelial-mesenchymal transition, which is a critical step driving tumor invasion and metastatic potential [108,109].

Indeed, in a comparison of organ-specific metabolic rates, resting SKM exhibited a rate of 13 kcal/kg/d, which is only 3.0–6.5% of the rate for heart, kidney, brain, and liver (440, 440, 240, and 200 kcal/kg/d, respectively) [102]. The incidence rate of kidney, brain, and liver cancer is 4.4–15.2, 3.5–5.7, and 8.5–8.6/100,000 persons, respectively [17,20]. In contrast, as reported in Section 1, the incidence rate of rhabdomyosarcoma in childhood is 4.5–4.9/1 million [13,14] and in adults is ~9.6/10 million; therefore, in adults, the incidence rate of rhabdomyosarcoma is only 2.2%, 2.7%, and 1.1% compared to that of kidney, brain, and liver cancers, respectively—these differences are even more pronounced when the incidence rate is expressed relative to tissue mass (listed in Table 1), with the value for rhabdomyosarcoma (incidence rate per 100,000 per g tissue mass) representing just 0.04%, 0.36%, and 0.48% of the values for kidney, brain, and liver cancers, respectively. Although further research is required to examine this phenomenon, with the exception of the heart (see Section 3.1.3), these numbers correspond to the differences in metabolic rates between resting SKM and other organs. This body of evidence indicates that the low metabolic rate of resting SKM may be one factor inhibiting cancer growth.

#### 3.1.5. Rich Blood Supply

Compared to most other tissues, SKM is highly vascularized and capillarized [110], and vasodilatory capacity can increase 100-fold [111] and perhaps as high as 800-fold [51]. Based on Poiseuille’s law, this indicates that SKM blood flow has the capacity to increase 1,000,000-fold (100^4^) or higher, which is far in excess of the 50- to 100-fold increase in SKM energy metabolism that occurs during contractile activity [101]. Vascular density, diameter, and capillary density are further increased with exercise training via angiogenesis [112]. The concomitant increased volume and velocity of blood flow in SKM during contractile activity, as well as the high intravascular shear forces, impede the ability of circulating tumor cells to extravasate to SKM and may disrupt the attachment of tumor cells to the intramuscular endothelium [51].

### 3.2. Immune-Related Factors

#### 3.2.1. Localized Immune Response

SKM contains resident immune cells, such as macrophages and lymphocytes [113]. Functionally, these immune cells are involved in the repair of damaged SKM [114]; however, they can also mount a localized immune response against cancer cells, limiting their ability to develop in or metastasize to SKM by preventing or limiting their proliferation [115]. Furthermore, the high vascularization and capillary density of SKM described in Section 3.1.5 result in a high ability to deliver anti-tumorigenic immune cells to SKM and the local microenvironment, further facilitating local anti-tumorigenic activity.

#### 3.2.2. Effect of Contractile Activity

In addition, SKM contractile activity initiates the conversion of pro-inflammatory M1 macrophages to anti-inflammatory M2 macrophages, resulting in reduced secretion of pro-angiogenic factors that support tumor growth. SKM contractile activity also prevents the conversion of anti-tumorigenic N1 neutrophils to pro-angiogenic N2 neutrophils that accelerate metastasis [65,116]. In a murine model of pancreatic cancer, the number and cytotoxic activity of CD8+ T-lymphocytes (cytotoxic T-cells) are also increased with SKM contractile activity, as is their accumulation in tumoral tissue [117,118,119]. Furthermore, SKM contractile activity mobilizes natural killer cells [120] and enhances intra-tumoral blood flow, thus increasing the delivery of anti-cancer immune cells to the tumor [119]. Furthermore, owing to increased metabolic activity during SKM contractile activity [121], SKM temperature increases [122,123]; this higher temperature may augment recruitment of natural killer cells and thus help to prevent cancer development or metastasis [124,125,126], potentially in an interleukin (IL)-6-dependent manner [124,125]. These changes in immune cell conversions and the number and activity of anti-tumorigenic immune cells may occur at a higher rate in local SKM, thus limiting the ability of tumors to grow in the SKM environment.

### 3.3. Skeletal Muscle Microenvironment

Whereas numerous studies have examined how SKM contractile activity can affect the tumor microenvironment in general [64,127,128,129,130], it is less clear how the specific SKM microenvironment may impact potential local tumorigenesis or infiltrating metastatic tumor cells. The possibilities of a physical barrier imposed by connective tissues and the shear stress in the connective tissues during muscle contraction comprise two components of the SKM microenvironment that may restrict tumorigenesis and are described in Section 3.1.2 and Section 3.1.3, respectively. In addition, other features of the SKM microenvironment related to cancer risk include ECM stiffness, pH, and oxygen tension. Myokine secretion, which may also be considered part of the microenvironment, is considered in Section 3.4.

#### 3.3.1. Extracellular Matrix Stiffness

One key feature of the tissue microenvironment that may affect cancer risk, progression, and metastasis is the ECM, including ECM stiffness [131]. This is a separate and distinct feature from the ECM stiffness of the tumor microenvironment, which changes and adapts over the course of tumor progression [132,133]. Although the influence of tissue ECM stiffness remains controversial [131], evidence indicates that cancerous cells from some origins preferentially adhere to and grow on ECM that exhibits less stiffness [134,135,136]. For instance, ovarian tumor cells preferentially invaded tissues with a lower degree of ECM stiffness [134]. Moreover, the formation of spheroid tumors was lower in three-dimensional fibrin gels with less stiffness [135]. Furthermore, when undifferentiated melanoma tumor-repopulating cells were grown on a fibrin gel with less stiffness, specific histone methylation occurred that limited colony growth [136]. Resting SKM exhibits >3-fold greater ECM stiffness compared to that of breast or dermal tissue [96], but the stiffness can increase 20-fold during tetanic contraction in an unloaded muscle [137]; stiffness likely increases even further in a loaded, fully contracted SKM. This high stiffness may prohibit the ability of metastasized cells to adhere to the SKM ECM and thus help to account for the rarity of SKM cancers.

#### 3.3.2. Lactic Acid and pH

Owing to the dependence on anaerobic glycolysis in tumors as explained by the Warburg effect [103], SKM-derived lactic acid at exercise intensities above the lactate threshold may disrupt cancer metabolism by further raising intracellular lactic acid in tumor cells, manifesting end-product inhibition, and thus consequently hindering tumor adhesion and tumor angiogenesis that are essential for tumor growth [51,138,139]. In addition to a direct inhibitory influence on metabolism, this effect may be partly mediated via suppressed expression of estrogen-related receptor-α [139].

Furthermore, systemic pH may decline with SKM contractile activity, particularly of an anaerobic nature, largely due to the accumulation of lactic acid [140]. Tumors tend to thrive in a lower pH environment, which contributes to tumor cell proliferation, migration, and invasion, as well as to intra-tumoral angiogenesis [141,142]. However, in creating this acidic extracellular pH, the cancer cells generate an alkaline intracellular pH, despite the higher lactate production [143]. SKM contractile activity-induced lactic acidosis may intracellularly acidify the cancer cells and thus disrupt tumor progression [144].

#### 3.3.3. Oxygen Tension

Hypoxia during early tumor development is an essential driver of intra-tumoral angiogenesis, which subsequently permits further growth and is essential for metastasis [145]. SKM contractile activity promotes increased intra-tumoral blood flow and oxygen level, thus interfering with intra-tumoral hypoxia [119,146] and consequently limiting tumor progression by interrupting energy metabolism.

#### 3.3.4. Oxidative Stress

ROS produced by increased SKM contractile activity may induce oxidative stress in tumor cells that have infiltrated SKM. Although ROS can be pro-tumorigenic, excess ROS results in oxidative stress, which interferes with the balance of oxidation and reduction and impairs tumor cell proliferation [147].

#### 3.3.5. Lipophilic Ligands of the Lipocalin Protein Family

Lipocalins are a protein family that serve as protease inhibitors during the cell cycle [148,149,150]. The role of lipocalins in cancer is controversial [149]; however, they bind to a wide variety of lipophilic ligands that regulate lipocalin activity [151]; these lipophilic ligands are present in SKM and display local anti-tumorigenic activity [152].

### 3.4. Myokines

Myokines are SKM-secreted humoral factors, and their discovery, beginning with IL-6 in 2000 [153], has defined SKM as an important endocrine organ [154,155] that engages in extensive and complex cross-talk with other organs, including cancerous tissues [64,130,155,156]. Over 600 myokines may be secreted [157], many of which are increased by SKM contractile activity [155,158]. Some myokines exert anti-tumorigenicity by limiting tumor cell growth, stimulating apoptosis, and inhibiting the epithelial-mesenchymal transition, migration, and invasion of cancer cells [64,65,78,130,156]; myokines may also positively modulate the anti-cancer immune response [130,159,160,161].

In addition to their endocrine function, many myokines also act in an autocrine and paracrine manner [158,162]. As the myokine concentration in the SKM microenvironment may be greater than in the circulation, the myokine-driven anti-tumorigenic effect may be greater in SKM and therefore contribute to the scarcity of SKM cancers. Oncostatin-M, irisin, secreted protein acidic and rich in cysteine (SPARC), decorin, brain-derived neurotrophic factor (BDNF), IL-6, IL-7, IL-10, and IL-15 are among the myokines manifesting anti-tumorigenic activity, and circulating levels of most of these increase with SKM contractile activity [130,156]. While myokines may also exert anti-tumorigenic influence via anti-inflammatory actions [163] or by counteracting systemic insulin resistance, hyperinsulinemia, and hyperlipidemia [164,165,166,167] via actions on other organs, this section of the review focuses on the effects of myokines on cancer cells [78], either directly or by means of immunological effects.

#### 3.4.1. Oncostatin-M

Multiple studies illustrate the anti-tumorigenic properties of oncostatin-M [72,156,168], which is a member of the IL-6 cytokine family [169]. For instance, oncostatin-M substantially inhibited the proliferation of mammary cancer cells [68,170], increasing caspase (a pro-apoptotic enzyme [171]) activity and apoptosis [68]. Furthermore, oncostatin-M decreased proliferation, migration, and invasion (and thus the epithelial-mesenchymal transition) of lung adenocarcinoma cells by activating STAT1, which increased E-cadherin expression [172]. As E-cadherin is a cell adhesion molecule and can act as a tumor suppressor [173], this suggests that oncostatin-M-mediated upregulation of E-cadherin may help to limit cancer metastasis. Notably, decreased E-cadherin expression is linked to a poor prognosis in patients with breast cancer [174]. In addition, in glioblastoma cells, oncostatin-M inhibited proliferation and increased expression of the cell cycle inhibitors p21 and p27^kip1^, with concomitantly decreased expression of Skp2, Cks1, and cyclin A [175], which help drive the cell cycle and are thus proto-oncogenic. Patients with prostate cancer exhibited increased serum oncostatin-M, but no changes occurred in other myokines; exercise-conditioned serum from these patients inhibited the growth of prostate cancer cells [80]. One mechanism by which oncostatin-M may inhibit tumor growth is by promoting the conversion of pro-inflammatory M1 macrophages to anti-inflammatory M2 macrophages and increasing neutrophil recruitment [176]. However, controversy exists regarding whether oncostatin-M may also exert pro-tumorigenic effects that depend on the specific tumor microenvironment [156].

#### 3.4.2. Irisin

Numerous lines of evidence also illustrate the anti-tumorigenic properties of irisin [156]. In mammary and lung cancer cells, irisin hindered proliferation and the epithelial-mesenchymal transition [67,177], but it did not affect the proliferation of mammary control cells [67]. Furthermore, in gastric carcinoma cells, irisin treatment or overexpression inhibited migration and invasion but did not affect proliferation [178]. Moreover, in ovarian cancer cells, irisin antagonized hypoxic cell signaling induced by hypoxia-inducible factor-1α [77], suggesting that irisin may interfere with the tumor response to hypoxia. In two of three ovarian cancer cell lines, it also reduced the expression of vascular endothelial growth factor (VEGF), which is essential for intra-tumoral angiogenesis [179]. Additionally, in pancreatic cancer cells, irisin inhibited proliferation, migration, and the epithelial-mesenchymal transition and induced cell cycle arrest at G1. It also upregulated E-cadherin protein expression and activated the AMPK-mTOR pathway [85]. Finally, in a mouse model of glioblastoma, irisin administration resulted in increased p21 expression in conjunction with cell cycle arrest at the G2/metaphase transition, inhibited invasion via upregulation of tissue factor pathway inhibitor-2, and reduced in vivo tumor size in mice by >85% [72], thus demonstrating the strong anti-tumorigenicity of this myokine.

#### 3.4.3. Secreted Protein: Acidic and Rich in Cysteine

SPARC, also referred to as osteonectin, induced apoptosis in ovarian cancer cells [180], mitigated proliferation of prostate cancer cells [181], inhibited proliferation and promoted apoptosis of implanted lung carcinoma and T-lymphoma cells in vivo in mice [182], and reduced tumor growth but accelerated invasion in the brain in a murine glioma model [183]. Additionally, SPARC overexpression in implanted gastric cancer cells interfered with the epithelial-mesenchymal transition and decreased cell viability and migratory ability; further, SPARC overexpression also inhibited VEGF expression [184,185]. Moreover, in a mouse model of colorectal cancer, wild-type control mice, but not SPARC knockout mice, demonstrated an exercise-induced reduction in the development of aberrant crypt foci (a hallmark of colorectal cancer [186]) and increased tumor apoptosis, indicating the essential involvement of SPARC in these anti-tumorigenic events. Furthermore, SPARC administration greatly attenuated tumorigenesis in the colon of SPARC knockout mice [70]. In addition, when lung carcinoma, T-lymphoma cells, or prostate cancer cells were subcutaneously injected in SPARC knockout mice, tumor growth was enhanced [181,182]. Tumor histology demonstrated altered ECM structure and decreased macrophage infiltration [182], suggesting that SPARC mediates interactions of tumor cells with the ECM. When SPARC knockout mice were crossed with TRAMP mice (a model of prostate adenocarcinoma), the genetic presence of SPARC resulted in a lower degree of tumor cell proliferation [181]. Mechanistically, SPARC inhibited tumor apoptosis in colorectal cancer via inhibition of Bcl-2, which is anti-apoptotic, as well as by upregulating autophagy [187]; SPARC also induced cell cycle arrest in medulloblastoma cells at the G2/metaphase transition via upregulation of p21 [188] and in prostate cancer by promoting expression of p21 and p27^kip1^ [181]; and finally, SPARC blocked intra-tumoral angiogenesis in pancreatic ductal carcinoma cells via direct binding to VEGF and platelet-derived growth factor [189]. However, as is the case with oncostatin-M, some evidence also suggests that SPARC may possess pro-tumorigenic activity [156].

#### 3.4.4. Decorin

Decorin overexpression in several culture models of cancer (colon, mammary, and squamous carcinoma) inhibited proliferation and promoted apoptosis [190,191,192,193], and decorin treatment of mammary cancer cells or an orthotopic mammary carcinoma model (either directly or via an adenoviral vector) reduced tumor growth by 70% and substantially curbed metastasis [193]. In addition, decorin treatment of endothelial cells reduced the angiogenic potential by downregulating VEGF [194]. Decorin knockout mice demonstrated intestinal tumorigenesis, with suppressed expression of p21 and p27^kip1^ [195]. Furthermore, in decorin knockout mice, colon cancer xenografts exhibited greater tumor weight and reduced E-cadherin expression in intestinal epithelial cells [190,195], whereas increasing decorin expression in colon cancer cells concomitantly upregulated E-cadherin expression [190]. Decorin may further act by downregulating the androgen receptor-phosphatidylinositol 3-kinase-Akt signaling pathway [196], the pro-tumorigenic miRNA miR-21 [197,198], transforming growth factor-β1 [198,199], cyclin D1 [199], and endothelial growth factor receptor activity [196,199,200], as well as by increasing p21 [199,200] and p53 [199].

#### 3.4.5. Brain-Derived Neurotrophic Factor

BDNF is most highly expressed in the brain [201], and the potential anti-tumorigenic role of SKM-derived BDNF remains elusive [156]. Although SKM-specific BDNF expression (both mRNA and protein) increases with exercise in mice [202] and humans, post-exercise plasma levels in humans remain unchanged [203]. This suggests that BDNF in SKM may act in an autocrine or paracrine manner. Evidence that BDNF exerts anti-tumorigenic activity includes the following: First, infusion of BDNF into a glioma-bearing mouse brain suppressed tumor growth and migration via a truncated TrkB.T1 receptor-dependent mechanism; it also suppressed intra-tumoral macrophage infiltration [204]. Second, hypothalamic BDNF overexpression in mice increased the number of CD8+ T-lymphocytes; this increase was essential for the anti-tumorigenic effect in an orthotropic model of melanoma [205]. Finally, hypothalamic BDNF overexpression in mice decreased melanoma tumor weight by about 80%, whereas blockade of hypothalamic BDNF expression via RNA interference mitigated the anti-tumorigenic intervention of environmental enrichment, which included a voluntary running wheel; this suggests that BDNF was responsible for the anti-tumorigenic effect [206]. Accordingly, SKM muscle-derived BDNF may act locally to inhibit tumor growth.

#### 3.4.6. Interleukin-6

IL-6 was the first myokine discovered [153] and is perhaps the most studied. The source of IL-6 appears to be important, with post-exercise increases in IL-6 exhibiting multiple anti-tumorigenic effects [78,207,208]. SKM is a primary source of exercise-induced increases in IL-6, although it has a short clearance rate [209]. Substantial evidence supports the anti-tumorigenic potential of SKM-derived IL-6, including its role in the SKM contractile activity-induced mitigation of cancer risk. For instance, anti-IL-6 antibodies at least partly eliminated the exercise-induced tumor suppression observed in mice [161]. Moreover, IL-6 inhibited proliferation of prostate cancer cells [210], colorectal cancer cells [211], and mammary cancer cells [212]; however, treatment protocols can alter the results [130]. In mammary cancer cells, IL-6 decreased proliferation via regulation of matrix metalloproteinases [212], thus IL-6 may directly regulate the tumor microenvironment; it also exhibits anti-tumorigenicity by increasing natural killer cell activity and intra-tumoral infiltration; these lymphocyte-mediated responses are necessary, but not sufficient, for anti-tumorigenicity [161], suggesting that other exercise-related humoral or additional factors are also involved.

#### 3.4.7. Interleukin-7

Strength training increased the expression of IL-7 mRNA in resting SKM [213]; IL-7 expression also increased following a single soccer match [214]. IL-7 demonstrates anti-tumorigenic potential [215,216], potentially through immunomodulation. IL-7 plays a role in the development and homeostasis of T- and B-lymphocytes as well as the immune response of natural killer cells and dendritic cells. Additionally, IL-7 promotes T- and B-lymphocyte precursor survival and proliferation, including the survival of both memory and naïve T-lymphocytes, and it contributes to the homeostasis of peripheral T-lymphocytes [217]. Nonetheless, the anti-tumorigenic role of SKM-derived IL-7 remains to be fully explored.

#### 3.4.8. Interleukin-10

IL-10 may also mediate anti-tumorigenic activity via immunomodulation by upregulating the number of anti-cancer immune cells and enhancing their cytotoxic activity [78]. In a mouse model of lymphoma, tumor growth was substantially reduced and memory CD8+ T-lymphocytes were increased in mice that were injected with IL-10 immediately following a booster vaccine relative to mice that were not inoculated and received no IL-10 [218]. Evidence also shows that IL-10 can inhibit angiogenesis, as indicated by a study employing a mouse model of melanoma, in which melanoma cells that expressed IL-10 via a transfected viral vector exhibited an attenuation of tumor growth and metastasis. When the melanoma cells lacking IL-10 expression were admixed with the transfected cells, tumor growth and metastasis remained diminished [219], confirming that the anti-tumorigenic activity was IL-10-dependent. The implanted tumors showed abated neovascularization, indicating that IL-10 may also exert anti-tumorigenicity by inhibiting intra-tumoral angiogenesis. This idea is further supported by IL-10-driven suppression of VEGF, tumor necrosis factor-α, and matrix metalloproteinase-9 expression [219].

#### 3.4.9. Interleukin-15

IL-15 regulates immune function by modulating natural killer cells and T-lymphocytes via stimulation of proliferation, differentiation, and maturation [159,160,220,221,222,223]. In a mouse model of luminal B mammary adenocarcinoma, fluorescently labeled IL-15 localized with CD8+ T-lymphocytes and natural killer cells found in the tumor and in the lymph nodes in the local lymphatic drainage [224]. Similar effects were also observed in a mouse model of metastatic triple-negative breast cancer (an aggressive, hormone-insensitive carcinoma [225]) treated with heterodimeric IL-15, which lowered the number of tumor cells in the blood and initiated a decline in tumor colonization of the lungs [226]. Additionally, IL-15 infusion into a glioma-bearing mouse brain suppressed tumor growth, which depended on intensified intra-tumoral infiltration of natural killer cells [204], and in a mouse model of pancreatic cancer, IL-15 limited tumor growth and improved survival; further, low-intensity treadmill exercise increased the number of circulating CD8+ T-lymphocytes as well as intra-tumoral infiltration; the observed effects depended on IL-15 [117].

Moreover, in 11 patients with CD52+ T-lymphocyte malignancies, 6 weeks of subcutaneous injection with human IL-15 resulted in increased serum levels of CD8+ lymphocytes and natural killer cells, as well as enhanced antibody-dependent cytotoxic activity of natural killer cells. Clinically, when IL-15 injection was administered in conjunction with alemtuzumab, two patients demonstrated a complete response, and two patients demonstrated a partial response [227]. In addition, exercise-induced IL-15 expression was associated with prolonged survival in patients with lung adenocarcinoma, colon adenocarcinoma, colorectal adenocarcinoma, esophageal carcinoma, skin cutaneous melanoma, uterine carcinosacrcoma, and rectal adenocarcinoma; survival outcomes also corresponded to intra-tumoral immune cell infiltration [228].

### 3.5. Insulin-like Growth Factor-1 and Associated Binding Proteins

Insulin-like growth factor (IGF-1) is a growth factor primarily secreted by the liver in response to growth hormone stimulation [229,230]. SKM is a secondary source of IGF-1 [230]. IGF-1 is found in slightly different forms that exhibit specific functions [231], and approximately 98% of circulating IGF-1 is bound to various binding proteins (IGFBP) [232]. SKM-derived IGF-1 is generally thought to manifest autocrine and paracrine effects in SKM growth and repair [231].

#### 3.5.1. Role of Insulin-like Growth Factor-1 and Binding Proteins in Cancer

The role of IGF-1 and its associated binding proteins in cancer has not been fully elucidated, particularly the role of SKM-derived IGF-1 and IGFBPs. Circulating IGF-1 [231] and IGFBP-2 [233] are typically considered to be pro-tumorigenic; IGF-1 exhibits pro-proliferative and anti-apoptotic properties [234,235,236,237], while IGFBP-2 enhances cancer cell survival, proliferation, invasion, and migration, as well as intra-tumoral angiogenesis [233]. In contrast, IGFBP-1 [238] and -3 [239,240,241] are generally considered to be anti-tumorigenic, although IGFBP-1 also displays some pro-tumorigenic effects [238]. IGFBP-1 mitigated invasion and metastasis in hepatocellular carcinoma [242], attenuated intra-tumoral angiogenesis in lymphoma [238], invoked a decrease in breast cancer cell proliferation [243,244,245], and induced apoptosis in prostate cancer cells [246]. Transfection of lung cancer cells with IGFBP-3 inhibited spheroid growth as well as growth of cells seeded in an ECM-based gel, and IGFBP-3 treatment of lung cancer cells stimulated apoptosis both in vitro and in vivo in mice [239], as well as inhibiting proliferation of non-small cell lung cancer cells [240]; it may act by preventing and even reversing the epithelial-mesenchymal transition [241].

#### 3.5.2. Association of Insulin-like Growth Factor-1 and Binding Proteins with Cancer

The association of circulating levels of IGF-1, IGFBPs, and insulin also supports their roles in cancer but is not always consistent with the molecular data. For instance, high serum IGF-1 and IGFBP-3 levels raise the risk of numerous cancers [238,247,248,249,250]; however, in a meta-analysis, IGFBP-3 decreased the risk of lung cancer [250]. Lower IGFBP-1 levels are generally, but not always, associated with poor outcomes in multiple cancers [238]. In contrast, higher IGFBP-2 indicates a poor prognosis [251,252]. Moreover, insulin, which is structurally similar to IGF-1, is also related to cancer risk and outcomes, with hyperinsulinemia linked to a greater risk of cancer and cancer-related death [253,254,255].

#### 3.5.3. Derivation of Insulin-like Growth Factor-1 and Binding Proteins from Skeletal Muscle

Decreased levels of insulin and IGF-1 are associated with exercise and may contribute to the decreased risk of cancer with exercise [167,256]. Although the liver is the primary source of circulating IGF-1, SKM may also be a significant source [230], especially following intense exercise [257]; however, directional changes in circulating IGF-1 levels following SKM contractile activity remain controversial. Regardless, evidence indicates that exercise may reduce IGF-1 levels. For example, in healthy men, low-intensity exercise modestly decreased IGF-1 and IGFBP-1 [258], while IGF-1 and IGFBP-3 declined following a 100-km walking race [259]. Moreover, 6 months of a moderate-intensity program in postmenopausal women resulted in lower IGF-1 and IGFBP-3 levels as well as lower insulin levels [260]. Serum IGF-1 levels were also attenuated in older participants following a 12-week progressive resistance exercise regimen [261]. A long-term exercise program (4–6 sessions/week for a mean of 14 years) also resulted in decreased circulating IGF-1 levels [262]. Furthermore, a mixed-exercise routine in adolescent men decreased circulating IGF-1 and IGFBP-2 levels [263]. Finally, 7 days of exercise combined with a prescribed diet in young men decreased SKM-derived plasma IGF-1 levels during the last 2 days [264]. Nevertheless, a disparity of results exists among studies utilizing different exercise programs [265,266,267,268,269]; this may indicate that SKM contractile activity-induced changes in IGF-1 and IGFBP levels may depend on factors such as age, pre-program fitness level, hormonal status, diet, sex, and the length, type, and intensity of the exercise intervention.

### 3.6. Studies Using Exercise-Conditioned Serum

#### 3.6.1. Animal Studies

Incubation of cancer cells in the presence of serum following acute exercise or exercise training decreases proliferation. According to one review [64], one important aspect collectively conveyed by these studies is that exercise-conditioned media impairs the ability of the cancer cells to seed properly and to form colonies. In one animal study, when mammary cancer cells were incubated in serum collected following 60 min of swimming exercise in mice, cell proliferation decreased by approximately half and was accompanied by a concomitant increase in caspase activity. Subsequent experiments revealed that increased SKM secretion of oncostatin-M was responsible for the inhibition [68].

#### 3.6.2. Human Studies

Numerous human studies also demonstrate the anti-tumorigenic effect of post-exercise serum on colorectal, prostate, breast, and pancreatic cancer cells. For instance, in colorectal cancer cells treated with serum collected from men with a high risk for colorectal cancer, proliferation modestly decreased, but was accompanied by a much larger rise in a marker of DNA damage. These results were mirrored when the cells were treated with IL-6, which was elevated in the post-exercise serum [211]. In addition, serum collected from survivors of colorectal cancer immediately following high-intensity exercise inhibited colorectal cancer cell proliferation [270].

Similarly, serum collected from men with metastatic cancer-resistant prostate cancer impaired the proliferation of prostate cancer cells. The serum contained increased levels of oncostatin-M and SPARC [79]. Moreover, prostate cancer cells displayed an approximately one-third inhibition of proliferation when treated with serum from healthy men collected following 60 min of bicycle activity, and injection of pooled serum into severe combined deficiency mice delayed tumor onset. Following a multiplex assay for serum growth factors, the authors determined that increased IGFBP-1 and decreased epidermal growth factor were candidates for the observed effect [271].

In another study, serum was collected from breast cancer survivors following a 2-h exercise session; both hormone-sensitive mammary cancer cells and hormone-insensitive triple-negative breast cancer cells treated with the serum exhibited decreased viability [272]. In contrast, serum collected from breast cancer survivors or a healthy control group following a 6-month exercise intervention showed no effect on mammary cell viability compared with serum collected pre-intervention, despite a higher level of IL-6 [272]. In addition, injection of exercise-conditioned serum into NMRI-Foxn1^nu^ immunodeficient mice blunted tumor cell viability and development; however, the responsible factors were epinephrine and norepinephrine [273], which are not derived from SKM.

In a unique study [84], serum was collected from patients with stage III/IV pancreatic cancer following repeated electrical stimulation of different SKMs over 12 weeks. When a pancreatic adenocarcinoma cell line was treated with the serum, it inhibited proliferation and promoted apoptosis. The serum was enriched in the myokines C-C motif chemokine ligand-4, C-X-C motif ligand-1, and IL-10; treatment of two different pancreatic cancer cell lines with a combination of the three myokines impaired migratory capacity and increased cell death by ~40–60%, as measured using DNA fragmentation. Notably, oncostatin-M increased SKM expression of C-X-C motif ligand-1 [176], which functions as a chemokine that attracts neutrophils [274]. These results illustrate that SKM contractile activity per se, rather than exercise, is sufficient to generate increases in anti-tumorigenic myokines.

### 3.7. Potential Fiber Type-Specific Effects

In general, skeletal myofibers are defined as type I and type II, depending on the type of myosin heavy chain expressed. Type I myofibers are optimized for fatigue resistance using aerobic metabolism and generally have a smaller cross-sectional area per fiber, whereas type II myofibers, with IIa and IIb subtypes, are generally more optimized for power and non-aerobic metabolism, including the phosphagen system and anaerobic glycolysis [275]. Myofiber type-specific myokine secretion has been reported [276,277,278], but whether a preponderance of type I or type II myofibers affects cancer risk is unknown. However, it is interesting to speculate that an abundance of either type I or type II myofibers may reduce cancer risk, and genetic models may help to provide evidence concerning this hypothesis. Numerous factors drive the expression of myofiber type [279]; here, only three models are provided for illustrative purposes.

#### 3.7.1. Effect of Type I Myofibers

In murine models, global overexpression of the transcriptional coactivator peroxisome proliferator-activated receptor-γ coactivator-1α (PGC-1α) drives abundant development of type I myofibers with increased mitochondrial density and capillarization [280]; in mice with SKM-specific expression of PGC-1α, circulating inflammatory markers decreased, and many deleterious age-related changes were ameliorated [281,282]. Notably, the anti-tumor molecule oncostatin-M, which is secreted by SKM, demonstrated an 8.3-fold elevation in the serum of mice with transgenic SKM-specific overexpression of PGC-1α [282]. This raises the possibility that a higher abundance of type I myofibers may be associated with decreased cancer risk; studies using genetic deletion of forkhead box O-1 (FOXO1) further support this possibility. FOXO1 knockout mice exhibited a decline in type I myofiber phenotype, as indicated by gene expression [283]. In addition, a PAX3-FOXO1 fusion protein hampered the differentiation of rhabdomyosarcoma cells [284], supporting a potential anti-tumorigenic connection among FOXO1, type I myofiber phenotype, and cancer.

#### 3.7.2. Effect of Type II Myofibers

In addition to its suppressive effect on type I myofibers, SKM-specific ablation of FOXO1 also increased the type II myofiber phenotype [285]; as mentioned in Section 3.7.1, rhabdomyosarcoma proliferation is inhibited by a PAX3-FOXO1 fusion protein [284]; accordingly, a decreased amount of type II myofibers was associated with a pro-tumorigenic effect, raising the possibility that the reverse may also be true. This prospect is further supported by evidence from male mice with SKM-specific deletions of Akt1/2, wherein the mice exhibit a decrease in type II myofibers. When these mice were exposed to a high-fat diet, they demonstrated earlier death due to a higher incidence of tumors [286]. While the data regarding the effect of myofiber type on cancer remain sparse and inconclusive, they support the hypothesis of an anti-tumorigenic effect by both an abundance of type I and type II myofibers.

## 4. Conclusions

SKM cancers are rare. This rarity may be due to (A) unique structural and cellular characteristics of SKM tissue, as follows: (1) SKM myofibers are amitotic; (2) SKM is organized in a manner that provides a physical barrier to potential invading cells; (3) SKM experiences frequent mechanical forces that may make it more difficult for invading cells to adhere to the tissue surfaces and may dislodge adherent cells; (4) SKM exhibits a low resting metabolic rate that is inversely connected to cancer risk; and (5) SKM demonstrates high vascularization and capillarization that can experience greatly increased blood flow and velocity, as well as high shear forces, which may make it difficult for invading cells to adhere to the intra-vascular endothelium and may dislodge adherent cells. (B) Immune-related factors may also contribute to the rarity of SKM, with an enhanced local anti-tumorigenic immune response that can be further escalated with SKM contractile activity. (C) Properties of the SKM microenvironment may also inhibit tumorigenesis or tumor invasion; these properties include (1) high stiffness of the SKM ECM, (2) higher lactic acid concentration and lower pH produced by anaerobic metabolism during SKM contractile activity that exceeds the lactate threshold, (3) higher oxygen tension imparted by SKM contractile activity, (4) SKM-induced increases in intra-tumoral oxidative stress, and (5) secretion of anti-tumorigenic lipophilic ligands. (D) A number of myokines in the SKM microenvironment may also decrease the likelihood of SKM cancers. The myokines oncostatin-M, irisin, SPARC, decorin, BDNF, and IL-6, -7, -10, and -15 exhibit anti-tumorigenic activity. (E) SKM contractile activity-induced decreases in IGF-1, IGFBP-2, and insulin levels and increased IGFBP-1 and -3 levels may also help to limit cancer development in SKM. (F) Furthermore, many of the anti-tumorigenic effects of SKM contractile activity have been demonstrated using conditioned media treatment of cultured cancer cells. (G) Finally, an abundance of type I or type II myofibers may display anti-tumorigenic manifestations.

Nonetheless, despite the inferential evidence supporting the role of these mechanisms in preventing SKM cancer, empirical evidence is lacking. To the author’s knowledge, none of these potential mechanisms has been directly tested. This is likely because research efforts are typically focused on understanding the mechanisms of common cancers rather than on understanding why a particular cancer is rare. In addition, some of these mechanisms may be difficult to confirm. Accordingly, the mechanisms proposed herein remain entirely hypothetical. Improved understanding will require studies with the primary aim of understanding how these mechanisms help to prevent SKM cancer. Moreover, individually, the relative contribution of each of these postulated mechanisms is unknown and remains a topic for further research, including how these factors may change depending on the frequency and type of SKM muscular contractile activity. However, collectively, these mechanisms help to explain why SKM, an organ comprising approximately one-half of body mass, exhibits the lowest rate of cancer per mass of any organ and avoids the cancer that is more prevalent in other organs. A better understanding of these mechanisms may be useful to identify improved prevention and treatment for cancer, including the role of regular SKM contractile activity in the form of exercise.

## Figures and Tables

**Figure 1 ijms-25-06480-f001:**
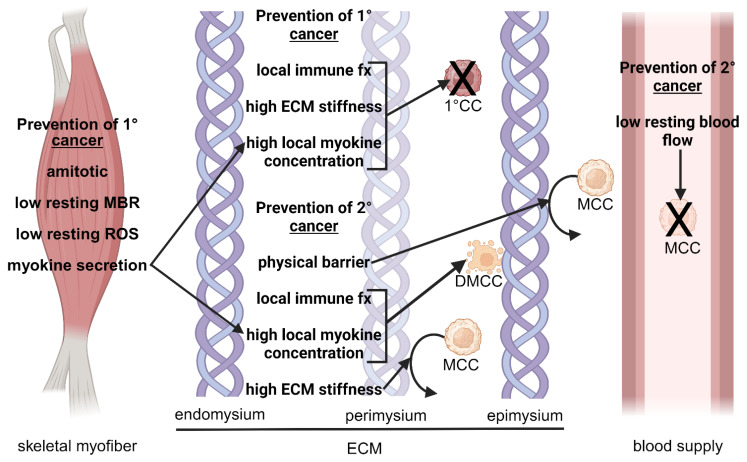
**Innate characteristics of skeletal muscle tissue that prevent primary and secondary cancer.** The diagram shows mechanisms that occur within the skeletal myofiber, the ECM, and the blood supply. Skeletal muscle ECM comprises the endomysium, which encases individual skeletal myofibers; the perimysium, which encloses bundles of myofibers (fascicles); and the epimysium, which surrounds an entire muscle. Although the figure shows the blood supply separately for clear illustrative purposes, the blood supply is found within each ECM component. The mechanisms work by directly inhibiting the formation of 1° cancer, causing apoptosis of existing MCC or otherwise interfering with tumor progression, blocking adherence of MCCs to the ECM, or preventing the spread of MCC to skeletal muscle. See the text for detailed mechanisms. Although the mechanisms in the ECM are shown at the perimysium-epimysium interface or superficial to the epimysium, they may occur anywhere within the ECM. Figure created using Biorender.com. Abbreviations: large X, prevention of formation of 1° CCs or prevention of movement of MCCs; 1°, primary; 2°, secondary; CC, cancer cell; DMCC, dying metastatic cancer cell; ECM, extracellular matrix; fx, function; MBR, metabolic rate; MCC, metastatic cancer cell; ROS, reactive oxygen species.

**Figure 2 ijms-25-06480-f002:**
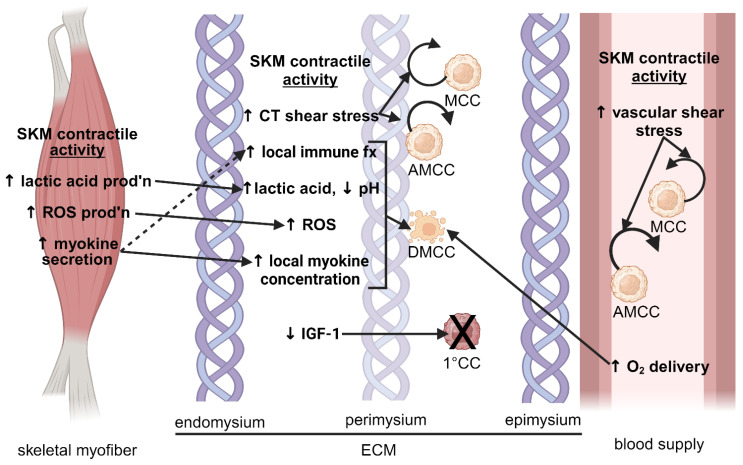
**Contractile activity-dependent mechanisms that prevent cancer in skeletal muscle.** The diagram shows contractile activity-dependent mechanisms that occur within the skeletal myofiber, the ECM, and the blood supply. SKM ECM comprises the endomysium, which encases individual skeletal myofibers; the perimysium, which encloses bundles of myofibers (fascicles); and the epimysium, which surrounds an entire muscle. Although the figure shows the blood supply separately for clear illustrative purposes, the blood supply is found within each ECM component. The mechanisms work by directly inhibiting the formation of 1° cancer, causing apoptosis of existing MCC or otherwise interfering with tumor progression, blocking adherence of MCCs to the ECM or the endothelium, or dislodging AMCCs attached to the ECM or endothelium. See the text for detailed mechanisms. Although the mechanisms in the ECM are shown at the perimysium-epimysium interface, they may occur anywhere within the ECM. Figure created using Biorender.com. Abbreviations: large X, prevention of formation of 1° CCs; 1° CC, primary cancer cell; AMCC, adherent metastatic cancer cell; CT, connective tissue; DMCC, dying metastatic cancer cell; ECM, extracellular matrix; fx, function; IGF-1, insulin-like growth factor-1; MCC, metastatic cancer cell; prod’n, production; ROS, reactive oxygen species; SKM, skeletal muscle.

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
