# Peer review of "Mechanisms Underlying the Rarity of Skeletal Muscle Cancers"

_ijms, 2024, doi:10.3390/ijms25126480_

Round 1

Reviewer 1 Report

Comments and Suggestions for Authors

In the paper entitled: Mechanisms underlying the rarity of skeletal muscle cancers, the author explores the mechanisms that help to explain the rarity of skeletal muscle cancers, including unique SKM architecture and function, which prohibits development of new cancer as well as negates potential metastasis to SKM. The methodologies employed are fitting and congruent with the specified objectives, combining preclinical data with clinical ones, while the conclusions resonate with the provided evidence and arguments. Nevertheless, I do have some suggestions for potential improvements.

1.    I would like the data in Table 1 to be accompanied by the corresponding bibliographic references.

2.    To make it easier to understand certain mechanisms and to establish correlations between clinical and preclinical data, I propose inserting 2-3 images in the main text.

3.    Addressing any limitations encountered during the study would be beneficial for providing a comprehensive understanding of its scope and potential constraints.

Reviewer 2 Report

Comments and Suggestions for Authors

The manuscript ijms-3024468 comprehensively reviews the potential mechanisms that might explain the resistance of skeletal muscle (SKM) to develop primary and secondary cancer. This is an original work relevant for the field and it may offer a novel perspective on how to look at cancer: instead of describing mechanisms supporting tumorigenesis, the author offers the opposite perspective by looking at why cancer is rarely developed by SKM. This knowledge may possibly lead to improved prevention and treatment strategies, further supporting the beneficial role of physical exercise based on in vitro and in vivo findings. This Reviewer is not aware of similar reviews in the scientific literature, which supports the originality of the current work and its role in filling such a gap.

Overall, the manuscript is well written, complete, and clear. The author cites and discusses a significant number of appropriate, up-to-date references relevant for the work. I have only a few suggestions that might help improve the quality of the work, as follows.

The manuscript is generally well-structured; however, sometimes sub-paragraphs are very short (e.g., 3.3.3. is five lines, 3.3.4. is four lines, 3.3.5. (incorrectly labelled as 3.3.4.) is, again, four lines. I wonder if bullet points could be more helpful instead.

The conclusion is mainly a summary of the discussed mechanisms, leaving only a little space to comments on future perspectives and knowledge gaps to be filled, which are therefore encouraged, if possible.

Line 554: please avoid citation of unpublished data.

Even though the manuscript is clear, I believe that one or two figures might be beneficial to summarize the main concepts discussed here and improve the overall readability.

Table 1 is a little tricky due to the extensive use of notes.
